# NeurQuRI: Neural Question Requirement Inspector for Answerability Prediction in Machine Reading Comprehension

**Seohyun Back**[1]**, Sai Chetan Chinthakindi**[1]**, Akhil Kedia**[1]**, Haejun Lee**[1] **and Jaegul Choo**[2]
Samsung Research, Seoul South Korea[1]     KAIST, Daejeon South Korea[2]
{scv.back,sai.chetan,akhil.kedia,haejun82.lee}@samsung.com
jchoo@kaist.ac.kr

## Abstract

Real-world question answering systems often retrieve potentially relevant documents to a given question through a keyword search, followed by a machine reading comprehension (MRC) step to find the exact answer from them. In this process, it is essential to properly determine whether an answer to the question exists in a given document. This task often becomes complicated when the question involves multiple different conditions or requirements which are to be met in the answer. For example, in a question "What was the projection of sea level increases in the fourth assessment report?", the answer should properly satisfy several conditions, such as "increases" (but not decreases) and "fourth" (but not third). To address this, we propose a neural question requirement inspection model called NeurQuRI that extracts a list of conditions from the question, each of which should be satisfied by the candidate answer generated by an MRC model. To check whether each condition is met, we propose a novel, attention-based loss function. We evaluate our approach on SQuAD 2.0, NewsQA and MS MARCO datasets by integrating the proposed module with various MRC models, demonstrating the consistent performance improvements across a wide range of existing methods.

## 1 Introduction

Machine reading comprehension (MRC), where a machine understands a given document and answers a question, is a challenging task, but it has a significant impact in real-world applications such as dialog systems. In practice, given a user-initiated question, potentially relevant paragraphs (often called contexts) are first retrieved from a search engine, which may or may not contain an actual answer. In this case, it is important for an MRC model (or in short, a reader) to be able to determine whether the retrieved context contains the answer before actually predicting the answer.

In most previous MRC tasks and datasets, such an answerability issue was out of scope as the provided context was guaranteed to contain an answer for a given question. Recently, a new dataset called SQuAD 2.0 (Rajpurkar et al., 2018) was released, containing instances with unanswerable questions for a given context, so that models can be properly trained to classify this case. Additionally, this dataset also contains information about plausible answers in the context when the question is unanswerable, which can be used to prevent our model from wrongly predicting it as an answer.

Previously, Liu et al. (2018) addressed the problem of classifying unanswerable cases by adding an auxiliary no-answer classifier to the last layer of the MRC model. Clark & Gardner (2018) tackled answerability classification through a joint softmax layer of the answerability score as well as the scores of all possible answer spans. Hu et al. (2019) attempted to verify the question against the sentence(s) containing the candidate answer. The answerability score from the verifier and the score from the reader were combined to generate the final score of having no answer.

However, these existing approaches do not pinpoint where the mismatch occurs between the question and the candidate answer in the unanswerable case, thus being prone to choosing a plausible but wrong answer. This task often becomes tricky when particular conditions from the question are not met. For example, in a question "What was the projection of sea level increases in the fourth

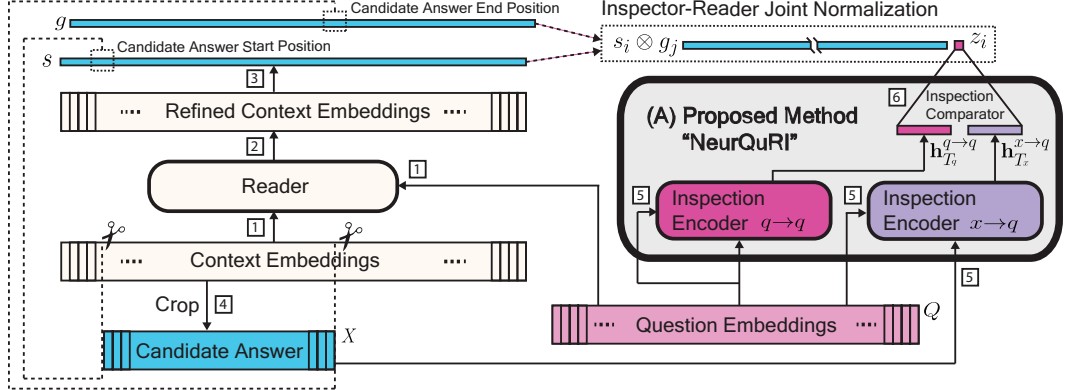

Figure 1: Overview of Neural Question Requirement Inspector (NeurQuRI). $\boxed{n}$ indicates the order of processes.

assessment report?", the answer should properly satisfy several conditions, such as "increases" (but not decreases) and "fourth" (but not third).

Motivated by this, we propose a novel neural inspector model that forms a list of conditions from the question, each of which should be satisfied by the candidate answer generated by the reader. To check whether each condition is met, we leverage and extend the idea proposed by Kiddon et al. (2016), which introduced a recurrent unit that records the used ingredients of cooking recipes by accumulating an attention mechanism during the generation of the recipe in a natural language text. They encourage the model to use all the ingredients by the end of the recipe text generation.

Extending this idea, we present a novel condition-checking module that determines whether the candidate answer satisfies all the conditions from the question. Furthermore, we propose a novel regularization method that can properly train our condition-checking model, leading to a correct candidate answer. Finally, we evaluate our proposed model on SQuAD 2.0, NewsQA (Trischler et al., 2017) and MS MARCO (Bajaj et al., 2016) datasets. Our experimental results show consistent improvements across a wide range of MRC models and also demonstrate the explainability of our model regarding which conditions of a given question are not met, or a reason why our model classified the question as unanswerable in a given context.

## 2 PROPOSED METHOD

This section discusses the details of our proposed method called Neural Question Requirement Inspector (NeurQuRI). As shown in Fig. 1-(A), NeurQuRI calculates answerability by taking a candidate answer and the question as input. To create the candidate answer, the reader takes a context (or paragraphs) and a question as input (Fig. 1-$\boxed{1}$), chooses the most probable candidate answer span (Fig. 1-$\boxed{2}$, $\boxed{3}$) and gives its contextualized word-level representation as input to NeurQuRI (Fig. 1-$\boxed{4}$). NeurQuRI then determines answerability by checking whether all conditions from the question are met by the candidate answer (Fig. 1-$\boxed{5}$, $\boxed{6}$). Intuitively, a wrong candidate answer will not satisfy at least one condition given by the question. To develop this idea, we propose a novel architecture for the inspector encoder as well as a condition satisfaction loss to properly train it.

### 2.1 NEURAL QUESTION REQUIREMENT INSPECTOR (NEURQURI)

In NeurQuRI, inspired by the idea of using an ingredient word as a condition in the checklist (Kiddon et al., 2016), each question word works as a condition to be satisfied by a candidate answer. Additionally, we use the question itself as the pseudo-answer that trivially contains all the words in the question and thus exemplifies an indication of full satisfaction during training. NeurQuRI creates and compares an inspection vector of the candidate answer with that of the question to check whether all the word-level meanings in the question are covered in the candidate answer.

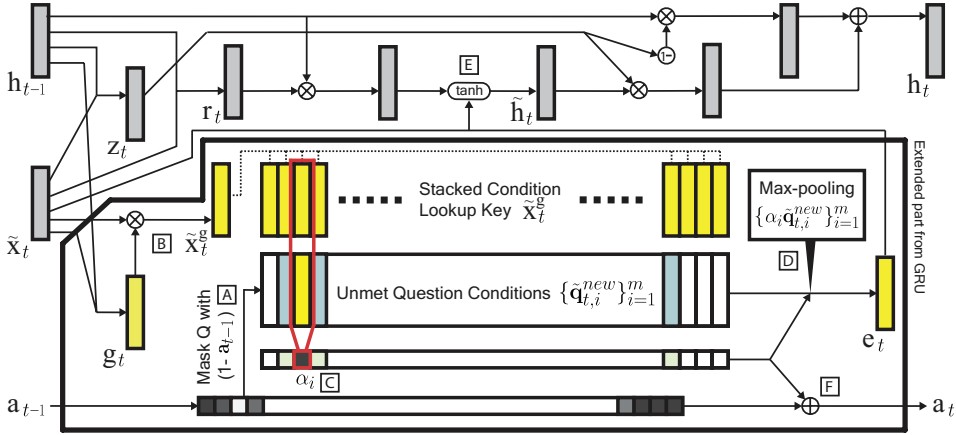

Figure 2: Architecture of our inspection encoder. Frame box corresponds to our extended part from the original gated recurrent unit (GRU).

NeurQuRI is largely composed of two parts: an inspection encoder and an inspection comparator. The inspection encoder encodes an input embedding sequence ($seq$), either a candidate answer or a question, into an inspection vector $\mathbf{h}^{seq \to q}$ by using the question's condition satisfaction with the input embedding sequence ($seq$). Using candidate answer ($x$) and question ($q$) as input sequence ($seq$), we calculate two inspection vectors, $\mathbf{h}^{x \to q}$ and $\mathbf{h}^{q \to q}$ (Fig. 1-⑤). Next, the inspection comparator compares these two vectors and finally computes an answerability score (Fig. 1-⑥).

**Inspection Encoder.** Let us denote a sequence of contextual embeddings of question words as $Q = \{\mathbf{q}_i\}_{i=1}^m \in \mathbb{R}^{m \times d_q}$ and that of candidate answer words as $X = \{\mathbf{x}_t\}_{t=1}^k \in \mathbb{R}^{k \times d_x}$, where $d_q$ and $d_x$ are the input dimensions of question and candidate answer words, respectively, and $m$ and $k$ are their respective sequence lengths. Given $Q$ and $X$, the inspection encoder generates (1) an inspection vector $\mathbf{h}$, which encodes information on the conditions given by $Q$ that are satisfied by $X$, and (2) the cumulative satisfaction score vector $\mathbf{a} \in \mathbb{R}^m$, each element of which indicates how much the condition corresponding to the question word is satisfied, as a value between zero and one. In detail, we extend a gated recurrent unit (GRU) (Cho et al., 2014), which sequentially takes a candidate answer word $\mathbf{x}_t$ at time $t$ and generates an inspection vector $\mathbf{h}_t$ and the cumulative satisfaction score vector $\mathbf{a}_t$, i.e.,

$$\mathbf{a}_t, \mathbf{h}_t = \text{InspectionEnc}(\mathbf{x}_t, \mathbf{a}_{t-1}, \mathbf{h}_{t-1}, Q),$$

where vectors $\mathbf{a}$ and $\mathbf{h}$ are initialized with zeros.

First, we linearly transform $Q$ and $X$ as $\tilde{Q} = \{\tilde{\mathbf{q}}_i\}_{i=1}^m \in \mathbb{R}^{m \times d}$ and $\tilde{X} = \{\tilde{\mathbf{x}}_t\}_{t=1}^k \in \mathbb{R}^{k \times d}$ to have the target dimension $d$ in common. We then calculate $\tilde{Q}_t^{new} = \{\tilde{\mathbf{q}}_{t,i}^{new}\}_{i=1}^m$, which represents unmet conditions by $X$ from $Q$ until time $t$, i.e.,

$$\tilde{Q}_t^{new} = ((1_m - \mathbf{a}_{t-1}) \otimes 1_d) \circ \tilde{Q} \in \mathbb{R}^{m \times d},$$

where $\circ$ indicates element-wise multiplication and $\otimes 1_d$ stacks the source $d$ times (Fig. 2-Ⓐ).

Second, we multiply the $t$-th word vector of a candidate answer, $\tilde{\mathbf{x}}_t$, with a gating vector $\mathbf{g}_t$ so that we can selectively ignore part of information contained in $\tilde{\mathbf{x}}_t$, yielding the vector $\tilde{\mathbf{x}}_t^g$ as

$$\mathbf{g}_t = \sigma(W_g \tilde{\mathbf{x}}_t + U_g \mathbf{h}_{t-1}) \in \mathbb{R}^d$$
$$\tilde{\mathbf{x}}_t^g = \mathbf{g}_t \circ \tilde{\mathbf{x}}_t \in \mathbb{R}^d,$$

where $W_g \in \mathbb{R}^{d \times d}$ and $U_g \in \mathbb{R}^{d \times d}$ are linear transformation matrices and $\sigma$ represents a sigmoid function (Fig. 2-Ⓑ). Ignoring such partial information prevents our model from considering the redundant information in the previous candidate answer words.

Third, we compute the satisfaction score at time step $t$, $\alpha_t$, by using $\tilde{\mathbf{x}}_t^g$ as a query vector and each question word (or condition) vector in $\tilde{Q}_t^{new}$, i.e.,

$$(\alpha_t)_i = \sigma(f_\alpha([\tilde{\mathbf{x}}_t^g; \tilde{\mathbf{q}}_{t,i}^{new}; \tilde{\mathbf{x}}_t^g - \tilde{\mathbf{q}}_{t,i}^{new}; \tilde{\mathbf{x}}_t^g \circ \tilde{\mathbf{q}}_{t,i}^{new}])),$$

where $f_\alpha$ is a fully connected unit with two hidden layers of dimensions $d/2$ and 1, respectively (Fig. 2-$\boxed{\text{C}}$). We used independent sigmoid outputs here instead of integrated softmax outputs to allow multiple high satisfaction scores for all met conditions. For example, word 'Lincoln' in the candidate answer can simultaneously satisfy the words 'Who' and 'President' in the question. We obtain the vector $\mathbf{e}_t$ as the information of the satisfied condition (Fig. 2-$\boxed{\text{D}}$) by max-pooling over $\tilde{Q}_t^{new}$ multiplied with the satisfaction score $\alpha$, i.e.,

$$\mathbf{e}_t = \text{max-pooling}(\{\alpha_i \tilde{\mathbf{q}}_{t,i}^{new}\}_{i=1}^m) \in \mathbb{R}^d.$$

Finally, we use $\mathbf{e}_t$ as an additional feature to compute the update vector in GRU, $\tilde{\mathbf{h}}_t$ (Fig. 2-$\boxed{\text{E}}$) and obtain the inspection vector $\mathbf{h}_t$ as

$$\mathbf{z}_t = \sigma(W_z \tilde{\mathbf{x}}_t + U_z \mathbf{h}_{t-1}) \in \mathbb{R}^d$$
$$\mathbf{r}_t = \sigma(W_r \tilde{\mathbf{x}}_t + U_r \mathbf{h}_{t-1}) \in \mathbb{R}^d$$
$$\tilde{\mathbf{h}}_t = \tanh(W_h \tilde{\mathbf{x}}_t + U_h(\mathbf{r}_t \circ \mathbf{h}_{t-1}) + V_h \mathbf{e}_t) \in \mathbb{R}^d$$
$$\mathbf{h}_t = (1 - \mathbf{z}_t) \circ \mathbf{h}_{t-1} + \mathbf{z}_t \circ \tilde{\mathbf{h}}_t \in \mathbb{R}^d,$$

where $\mathbf{z}_t$ and $\mathbf{r}_t$ are an update and a reset gate, respectively, and $W_z$, $U_z$, $W_r$, $U_r$, $W_h$, $U_h$, $V_h$ $\in \mathbb{R}^{d \times d}$. We also update the cumulative satisfaction score vector $\mathbf{a}_t$ (Fig. 2-$\boxed{\text{F}}$), with the current satisfaction score vector $\alpha$ as

$$\mathbf{a}_t = \min(\mathbf{a}_{t-1} + \alpha, \mathbf{1}^m) \in \mathbb{R}^m,$$

where $\mathbf{1}^m$ is an $m$ dimensional ones vector so that each element of $\mathbf{a}_t$ is clipped between 0 and 1. $\mathbf{a}_t$ represents how much its corresponding question word as a condition is satisfied by the candidate answer.

**Inspection Comparator.** The inspection comparator compares the inspection vector of the candidate answer with the inspection vector of the question itself to check whether all the word-level meanings in the question are involved in the candidate answer. We use the question itself as the pseudo-answer that trivially contains all the words in the question. To be specific, we compare the 'candidate-answer-to-question' inspection vector $\mathbf{h}^{x \to q}$ with the 'question-to-question' inspection vector $\mathbf{h}^{q \to q}$, where the latter can be considered as a fully satisfied reference to the question. We compare these vectors using a fully-connected layer to generate an answerability score.

First, we calculate the inspection vector $\mathbf{h}_t$ and the question's satisfaction score $\mathbf{a}_t$ with respect to the question representation $Q$, i.e.,

$$\mathbf{a}_t^{x \to q}, \mathbf{h}_t^{x \to q} = \text{InspectionEnc}(\mathbf{x}_t, \mathbf{a}_{t-1}^{x \to q}, \mathbf{h}_{t-1}^{x \to q}, Q)$$
$$\mathbf{a}_t^{q \to q}, \mathbf{h}_t^{q \to q} = \text{InspectionEnc}(\mathbf{q}_t, \mathbf{a}_{t-1}^{q \to q}, \mathbf{h}_{t-1}^{q \to q}, Q).$$

Afterwards, we compute the answerability score $z_i$ by combining them, i.e.,

$$z_i = f_\beta([\mathbf{h}_{T_x}^{x \to q}; \mathbf{h}_{T_q}^{q \to q}; \mathbf{h}_{T_x}^{x \to q} - \mathbf{h}_{T_q}^{q \to q}; \mathbf{h}_{T_x}^{x \to q} \circ \mathbf{h}_{T_q}^{q \to q}),$$

where $T_x$ and $T_q$ indicate the last time steps of the two sequences, respectively, and $f_\beta$ is a fully connected unit with two hidden layers that have have dimensions as $d/2$ and 1, respectively.

## 2.2 Loss Function for Training NeurQuRI

MRC datasets such as SQuAD 2.0 generally contains the answerability label, which we call $\phi$, for a pair of a given question and a context (e.g., $\phi = 1$ means unanswerable). However, NeurQuRI predicts the answerability given as input an arbitrary, candidate answer span, which is contextualized by the given context, as well as the question. In this new setting, the label $\phi_d$ for such input should be ideally set as answerable ($\phi_d = 0$), only if (1) the given question is answerable from the context ($\phi = 0$) and (2) the candidate answer span exactly matches the ground-truth answer span.

However, we found it an overly strict condition detrimental in the overall accuracy. For example, given a ground-truth answer span "The American president, Abraham Lincoln", the candidate answer span "president, Abraham Lincoln" should perhaps be treated as properly answering the given

question. Thus, by relaxing the above strategy, we consider the candidate answer span being answerable as long as the candidate answer span contains at least a particular fraction of ground truth answer words, where such a fraction can be viewed as a recall measure, i.e.,

$$\phi_d = \begin{cases} 1, & \text{if } \phi = 1 \\ 1, & \text{if } \phi = 0 \text{ and } \text{Recall}(x_{span}, a_{span}) \leq \eta \\ 0, & \text{if } \phi = 0 \text{ and } \text{Recall}(x_{span}, a_{span}) > \eta, \end{cases}$$

where $\eta$ is a threshold of the minimum recall score to be answerable. We set $\eta$ as 0.5 in our experiments.

**Answerability Classification Loss.** To calculate the answerability classification loss $\mathbb{L}_i$, we use a cross-entropy loss between NeurQuRI's answerability score $z_i$ and dynamically modified ground truth $\phi_d$, i.e.,

$$\mathbb{L}_i = -\phi_d \log(\sigma(z_i)) - (1 - \phi_d) \log(1 - \sigma(z_i)).$$

**Satisfaction Score Loss.** We intend NeurQuRI to work as a checklist over conditions given in the question. This loss is designed to make a candidate answer fail to satisfy at least one condition of a question in unanswerable cases, i.e., to have at least one small value in the 'candidate-answer-to-question' satisfaction score vector $\mathbf{a}_T^{x \to q}$, but otherwise to have all high values. Also, all scores in the 'question-to-question' satisfaction score vector $\mathbf{a}_T^{q \to q}$ are enforced to have all high values because the question itself should satisfy all conditions of the question, by adding this to the loss with the weight $\gamma$. Afterwards, we calculate the final satisfaction score loss $\mathbb{L}_a$ as

$$\mathbb{L}_a = -\gamma \log(\min(\mathbf{a}_T^{q \to q})) - \phi_d \log(1 - \min(\mathbf{a}_T^{x \to q})) - (1 - \phi_d) \log(\min(\mathbf{a}_T^{x \to q})),$$

where we set $\gamma$ as 0.5 in our experiments.

**Inspector-Reader Joint Normalization.** Similar to Clark & Gardner (2018), we jointly normalize the answerability score, $\mathbb{L}_j$, of NeurQuRI and the span prediction score from the reader as

$$\mathbb{L}_j = -\log\left(\frac{\phi_d e^{z_i} + (1 - \phi_d) e^{s_a + g_a}}{e^{z_i} + \sum_{i=1}^{n} \sum_{j=1}^{n} e^{s_i + g_j}}\right),$$

where $s_i + g_j$ indicates the summation of the prediction score of the answer span from the start and the end token indices, $i$ and $j$, respectively, in the context with length $n$, and $s_a$ and $g_a$ indicate scores for the position of the ground truth start and end indices, respectively. In this manner, NeurQuRI's answerability score can overcome a wrongly predicted candidate answer from the reader while penalizing the reader. Finally, the total loss $\mathbb{L}_{total}$ is obtained as

$$\mathbb{L}_{total} = \lambda_i \mathbb{L}_i + \lambda_a \mathbb{L}_a + \lambda_j \mathbb{L}_j, \tag{1}$$

where $\lambda_i$, $\lambda_a$ and $\lambda_j$ are hyperparameters.

## 3  EXPERIMENTAL SETUP

**Reader.** The reader can be any MRC model. We use three popular publicly available reader models: BERT[1] (Devlin et al., 2019), DocQA[2] (Clark & Gardner, 2018), and QANet[3] (Yu et al., 2018). As depicted in Fig. 1-[n], for each training iteration, the reader first extracts a candidate answer and then NeurQuRI calculates its loss based on the extracted candidate answer followed by simultaneously updating NeurQuRI and the reader.

Additionally, we apply two auxiliary loss functions for the reader from previous work to improve candidate answer prediction. We utilize the loss for normalizing span distribution by an empty word (Liu et al., 2018). We also utilize the independent span loss for plausible answer (Hu et al., 2019) to boost the reader's candidate answer selection in unanswerable cases for SQuAD 2.0. Details on the auxiliary losses can be found in our supplemental material.

---

[1] https://github.com/google-research/bert
[2] https://github.com/allenai/document-qa
[3] https://github.com/NLPLearn/QANet

Table 1: Results reported on SQuAD 2.0. All the results are from their own publications, except for those with a dagger($^\dagger$), which are reproduced. The symbols on the left indicate the corresponding comparison group which are explained in Section 4.

| | Model | Dev Set | | Test Set | |
|---|---|---|---|---|---|
| | | EM | F1 | EM | F1 |
| | BERT (Large) + NeurQuRI (Batch size 24, ensemble) | 81.0 | 83.9 | 82.8 | 85.7 |
| ♢ | BERT (Large) + NeurQuRI (Batch size 24) | 80.0 | 83.1 | 81.3 | 84.3 |
| ♣ | BERT (Large) + NeurQuRI (Batch size 6) | 80.0 | 82.9 | 80.6 | 83.4 |
| | BERT (Large) + SG-Net Verifier (Batch size 24) | 79.6$^\dagger$ | 82.3$^\dagger$ | - | - |
| ♢ | BERT (Large) (Devlin et al., 2019) (Batch size 24) | 78.7$^\dagger$ | 81.8$^\dagger$ | 80.0 | 83.1 |
| ♣ | BERT (Large) (Devlin et al., 2019) (Batch size 6) | 78.0$^\dagger$ | 80.9$^\dagger$ | - | - |
| ♠ | DocQA (ELMo) + NeurQuRI | 70.5 | 73.8 | 68.8 | 71.7 |
| | DocQA (ELMo) + Answer Verifier (Hu et al., 2019) | 68.0 | 70.7 | - | - |
| | DocQA (ELMo) + SG-Net Verifier (Zhang et al., 2019) | 67.8$^\dagger$ | 70.7$^\dagger$ | - | - |
| ♠ | DocQA (ELMo) + Joint No-answer (Rajpurkar et al., 2018) | 65.1 | 67.6 | 63.4 | 66.3 |
| ♡ | QANet + NeurQuRI | 65.3 | 68.9 | - | - |
| | QANet + SG-Net Verifier (Zhang et al., 2019) | 64.1$^\dagger$ | 67.6$^\dagger$ | - | - |
| ♡ | QANet + Joint No-answer (Rajpurkar et al., 2018) | 63.6$^\dagger$ | 66.7$^\dagger$ | - | - |
| | SLQA+ (Wang et al., 2018) | - | - | 71.5 | 74.4 |
| | RMR + Answer Verifier (Hu et al., 2019) | 72.3 | 74.8 | 71.7 | 74.2 |
| | Unet (Sun et al., 2018) | 70.3 | 74.0 | 69.2 | 72.6 |
| | SAN (Liu et al., 2018) | 69.3 | 72.2 | 68.7 | 71.4 |
| | DocQA + Joint No-answer (Rajpurkar et al., 2018) | 61.9 | 64.8 | 59.3 | 62.3 |
| | BiDAF + No Answer (Rajpurkar et al., 2018) | 59.8 | 62.6 | 59.2 | 62.1 |
| | Human Performance | 86.3 | 89.0 | 86.9 | 89.5 |

Table 2: Results of applying NeurQuRI to BERT reader on the NewsQA test set & MS MARCO dev set. The scores with a dagger($^\dagger$) are reproduced. ACC indicates answerability classification accuracy.

| Model | NewsQA | | | MS MARCO | | |
|---|---|---|---|---|---|---|
| | EM | F1 | ACC | EM | F1 | ACC |
| BERT (Large) + NeurQuRI (Batch size 24) | 48.2 | 59.5 | 81.3 | 45.7 | 54.6 | 69.9 |
| BERT (Large) (Batch size 24) | 46.5$^\dagger$ | 56.7$^\dagger$ | 77.9$^\dagger$ | 45.5$^\dagger$ | 53.4$^\dagger$ | 67.7$^\dagger$ |

**Benchmark Dataset.** We evaluate our model on SQuAD 2.0,[4] which contains unanswerable questions generated by crowd workers for the same paragraphs in SQuAD 1.1 (Rajpurkar et al., 2016). The training dataset contains 87K answerable and 43K unanswerable questions. The unanswerable questions are created such that a particular span in a context exists as a plausible but incorrect answer. We also evaluate our model on NewsQA (Trischler et al., 2017) which is a question answering dataset on paragraphs of news articles that tend to be longer than SQuAD. The dataset has 20K unanswerable questions among 97K questions. Additionally, we evaluate our model on MS MARCO (Bajaj et al., 2016) which has questions collected through Bing search engine and has answers with free-form text. The dataset has 305K unanswerable questions among 808K questions. Since the official evaluation of MS MARCO uses BLEU-4 score and completely ignores unanswerable cases, we report the results by following the same evaluation procedure as performed on SQuAD 2.0. We evaluate the performance on these datasets using standard metrics, EM and F1.

**Implementation Details.** We use the pre-trained Large BERT model using all the official hyperparameters and all hidden dimensions $d$ set as 1024. In particular, we evaluate our approach combined with BERT on SQuAD 2.0 leaderboard with the batch size of 24, but our ablation studies in Table 3 are performed with the batch size of 6 due to the limited GPU memory. In DocQA and QANet, we utilize 'Joint No-answer' for an answerability classification baseline as is used in SQuAD 2.0 (Rajpurkar et al., 2018). For these readers, we use GloVe 300d (Pennington et al., 2014) for word em-

---

[4] https://rajpurkar.github.io/SQuAD-explorer

beddings, along with the batch size of 24 and all hidden vector dimensions $d$ set as 200. In DocQA, we use ELMo (Peters et al., 2018) for contextualized embeddings. For comparison with existing verifying layer from Zhang et al. (2019), we reproduced the verifying layer on the all three readers.

For training our model, the hyperparameters $(\lambda_i, \lambda_a, \lambda_j)$ in Eq. equation 1 are set as (1.0, 1.0, 1.0), respectively. We choose these hyperparameters based on the performance of 'BERT (Large) + NeurQuRI' for the dev set. During inference, we compute the final score of the answerability by jointly normalizing $z_i$ with the span prediction scores.

After applying NeurQuRI to BERT (Large), the number of parameters increased by 9% (340M to 373M), the computation cost increased by 13% (706B to 803B), and the training speed decreased by 24% (1.6iter/sec to 1.2iter/sec).

## 4 QUANTITATIVE ANALYSIS

**Main Result.** As shown in Table 1, when using BERT (Large), DocQA (with ELMo), and QANet (without ELMo) as three different readers representing a high-, a medium- and a low-performance readers, respectively, our NeurQuRI consistently improves the performance on all the cases from the baseline on the SQuAD 2.0. For QANet (Table 1-♡), our approach achieves the F1 score of 68.9 (vs. 66.7) on the dev set. For DocQA (Table 1-♠), our approach outperforms the baseline, achieving the F1 score of 73.8 (vs. 67.6) on the dev set, and it achieves the F1 score of 71.7 (vs. 66.3) on the test set. For a recently proposed reader model called BERT (Table 1-♣), our approach with the batch size of 6 achieves the F1 score of 82.9 (vs. 80.9) on the dev set. The same model with the batch size set as 24 (Table 1-◇) achieves the F1 score of 83.1 (vs. 81.8) on the dev set, and it obtains the F1 score of 84.3 (vs. 83.1) on the test set. Moreover, including the verifying networks from Zhang et al. (2019) and from Hu et al. (2019), NeurQuRI also outperforms existing verifying networks based on the same reader in all the cases.

We also evaluate NeurQuRI on NewsQA and MS MARCO datasets by combining it with BERT reader. As shown in Table 2, NeurQuRI consistently improves all the evaluation metrics including the answerability classification accuracy. In detail, NeurQuRI improves the F1 scores by 2.8 and 1.2 for NewsQA and MS MARCO, respectively. The improvement of the EM score on MS MARCO is relatively small, e.g., +0.2. We conjecture that this is because the ground truth answer of MS MARCO is free-form text which is not ideal to the span prediction output of a typical BERT reader.

**Ablation Study on Different Loss Terms.** As shown in the 'Dev Set' column of Table 3-(a), we perform an ablation study with different combinations of the loss terms of NeurQuRI on the dev set in SQuAD 2.0. The answerability classification loss $\mathbb{L}_i$, the condition satisfaction score loss $\mathbb{L}_a$, and the inspector-reader joint normalization $\mathbb{L}_j$ all increase the performance consistently for all three reader models. Although not shown in the table, the performance using only $\mathbb{L}_j$ with BERT reader achieves an F1 score of 82.0, and the performance using $\mathbb{L}_a + \mathbb{L}_j$ with BERT reader obtains an F1 score of 82.1, which are low compared to other results including $\mathbb{L}_i$.

**Ablation Study on Excluding Stop Words.** We explore whether we should include stop words, which are often semantically less meaningful, as part of our conditions to consider in a given question. As shown in Table 3-(b), we evaluate 'BERT (Large) + NeurQuRI' model by excluding stop words from the question. In detail, we masked out stop word embedding vectors from the question embedding matrix before passing it to NeurQuRI. Excluding stop words actually decreases the performance of the EM/F1 score and the answerability classification accuracy. We conjecture the reason is because the stop words actually contain nontrivial information to determine the answerability of a given question. For example, Word 'is' indicates the present tense while 'was' does the past one. Word 'above' specifies clearly different relations from 'below'. In this respect, we included all the words of a given question as condition words for NeurQuRI to consider.

**Effects of Modification of Answerability Label.** As shown in Table 4, training with the modified answerability label ($\phi_d$) consistently boosts the performance, compared to the case of training with the original ground-truth label of answerability. In particular, the performance margin is bigger for the model with relatively lower F1/EM scores, e.g., the F1 margin of +1.4 for 'QANet+NeurQuRI, compared to that of 0.5 for 'BERT+NeruQuRI'. We conjecture that the modified answerability label is more effective for training low-performance readers that are more likely to give an incorrect answer, which should be predicted as unanswerable in NeurQuRI.

Table 3: Ablation studies on NeurQuRI. The results are obtained from the development set in SQuAD 2.0 using BERT (Large), DocQA (ELMo), and QANet as readers.

(a) Ablation study on the proposed losses.

| Reader | $\mathbb{L}_{total} +=$ | | | Dev Set | |
|---|---|---|---|---|---|
| | $\mathbb{L}_i$ | $\mathbb{L}_a$ | $\mathbb{L}_j$ | EM | F1 |
| Baseline | | | | 78.0 | 80.9 |
| | ✓ | | | 79.7 | 82.7 |
| BERT | ✓ | ✓ | | 79.9 | 82.8 |
| | ✓ | ✓ | ✓ | 80.0 | 82.9 |
| Baseline | | | | 65.1 | 67.6 |
| | ✓ | | | 69.5 | 72.4 |
| DocQA | ✓ | ✓ | | 69.6 | 72.3 |
| | ✓ | ✓ | ✓ | 70.5 | 73.8 |
| Baseline | | | | 63.6 | 66.7 |
| | ✓ | | | 64.1 | 68.3 |
| QANet | ✓ | ✓ | | 64.2 | 68.7 |
| | ✓ | ✓ | ✓ | 65.3 | 68.9 |

(b) Performance comparison between the cases with and without stop words in the question before passing it as an input to NeurQuRI. ACC indicates answerability classification accuracy.

| BERT + NeurQuRI | Dev Set | | |
|---|---|---|---|
| | EM | F1 | ACC |
| | 80.0 | 82.9 | 85.6 |
| - Stop words | 79.1 | 82.0 | 84.7 |

Table 4: Comparison on the EM, F1 scores of models between training with the ground-truth of answerability label ($\phi$) and training with the modified answerability label ($\phi_d$) which is explained in Section 2.2. The results are obtained from the development set in SQuAD 2.0.

| Reader + NeurQuRI | Training with $\phi$ | | Training with $\phi_d$ | | |
|---|---|---|---|---|---|
| | EM | F1 | EM | F1 | F1 margin |
| BERT (Large) + NeurQuRI (Batch size 24) | 79.7 | 82.6 | 80.0 | 83.1 | +0.5 |
| DocQA (ELMo) + NeurQuRI | 69.7 | 72.8 | 70.5 | 73.8 | +1.0 |
| QANet + NeurQuRI | 64.6 | 67.5 | 65.3 | 68.9 | +1.4 |

**Comparison with bi-LSTMs** We compare NeurQuRI against other basic DNN layers for answerability classification. We replace NeurQuRI with the simple LSTM (Hochreiter & Schmidhuber, 1997) layer for showing effectiveness of checklist mechanism. As same as applying NeurQuRI to the reader, we use BERT (Large) as the reader and give the question and candidate answer's contextualized word-level representations as input to the bi-LSTM layer. The final hidden-state vector and the cell-state vector of the LSTM module are concatenated and passed through a feed-forward layer to get the answerability score. The number of hidden units of the LSTM module are chosen so as to keep the same number of extra parameters as that of NeurQuRI's (33M). All hyperparameters for the reader are unchanged. As shown in Table 5, NeurQuRI outperforms bi-LSTM on all metrics, illustrating that the proposed method is superior to traditional approaches given the same amount of increases in parameters.

## 5 QUALITATIVE ANALYSIS

Fig. 3 presents three unanswerable examples from the dev set in SQuAD 2.0, demonstrating the effectiveness of NeurQuRI in explaining why a question is unanswerable. For these examples, we used the BERT (Large) as the reader. In these examples, Those words (conditions) rendering the questions unanswerable exhibits a low question satisfaction score $\mathbf{a}_T^{x \to q}$. For each question, we also prepare for an answerable question by making the smallest changes with the reader's candidate answer as its answer.

In the first example containing the question "When did Hutton die?", the context has no information about death, and the candidate answer "1795" is the date of "publishing". Hence the $\mathbf{a}_T^{x \to q}$ score for "die" is shown to be low (implying an unmet condition). However, using the modified question, with "die" replaced with "publish", all scores are shown to be high, implying each condition from the question is fully satisfied. Similarly, the candidate answers "11,700 years ago" and "for two bosons" do not satisfy the question's conditions "end" and "occasionally" in the second and the third exam-

Table 5: Comparison on the EM, F1 and the answerable classification accuracy between bi-LSTM and NeurQuRI based on the BERT (Large) reader. The results are obtained from the dev set in SQuAD 2.0. The size of parameters are kept same on both answerable classification module (33M).

| Metric | BERT + bi-LSTM | BERT + NeurQuRI | BERT + bi-LSTM | BERT + NeurQuRI |
|---|---|---|---|---|
| Batchsize | 6 | 6 | 24 | 24 |
| EM | 79.5 | 80.0 | 79.6 | 80.0 |
| F1 | 82.3 | 82.9 | 82.5 | 83.1 |
| ACC | 82.9 | 85.6 | 83.3 | 86.5 |

James Hutton is often viewed as the first modern geologist. In 1785 he presented a paper entitled Theory of the Earth to the Royal Society of Edinburgh. In his paper, he explained his theory that the Earth must be much older ... (omit) ... which in turn were raised up to become dry land. Hutton published a two volume version of his ideas in 1795 (Vol. 1, Vol. 2).

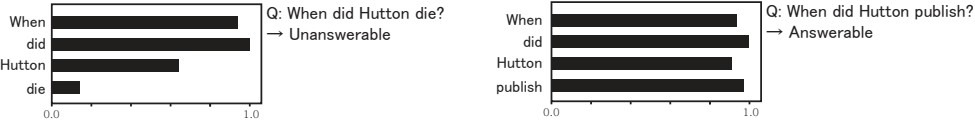

At the begin of the Holocene (~11,700 years ago), the Rhine occupied its Late Glacial valley. As a meandering river, it reworked its iceage braidplain. As sea level continued to rise in the Nethelands, the formation of the Holocene Rhine Meuse delta began (~8,000 years ago). ... (omit) ... the coastal marine dynamics, such as barrier and tidal inlet formations.

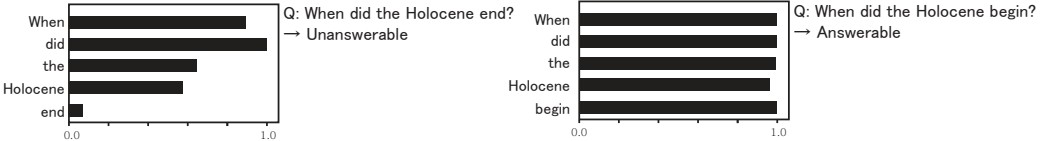

However, already in quantum mechanics there is one "caveat", namely the particles acting onto each other do not only possess the spatial, ... (omit) ... Thus in the case of two fermions there is a strictly negative correlation between spatial and spin variables, whereas for two bosons (e.g. quanta of electromagnetic waves, photons) the correlation is strictly positive.

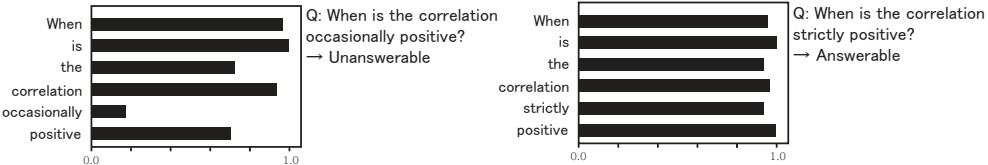

Figure 3: Prediction examples of our method, BERT (Large) + NeurQuRI, from the development set in SQuAD 2.0 and its satisfaction score vector $\mathbf{a}_T^{x \to q}$ over question words by the candidate answer. A colored text indicates the candidate answer predicted by the reader.

ples, respectively, as properly shown in our model. Correspondingly, after replacing these words, all condition satisfaction scores become high. These examples clearly demonstrate that NeurQuRI can reason why our model classified a question as unanswerable in a given context. Additional examples including failure cases can be found in our supplemental material.

## 6 RELATED WORK

**Reader Model.** Given a context guaranteed to have an answer to a question, the state-of-the-art machine reading comprehension now matches or even surpasses human performance. Wang et al. (2017) and Clark & Gardner (2018) achieved high performance by using a self-attention mechanism combined with recurrent neural networks. Yu et al. (2018) and Back et al. (2018) improved the performance by leveraging self-attention in each context-encoding neural networks block. Hu et al. (2018) predicted an answer span with a memory-based answer pointer using a semantic fusion unit across multiple hops, and Devlin et al. (2019) recently boosted the performance significantly by stacking the self-attention blocks proposed in the machine translation model (Vaswani et al., 2017). These models are widely utilized as the baseline for numerous MRC models.

**Answerability Prediction Model.** Recently, Liu et al. (2018) attempted to solve predicting answerability by appending an empty word token to the context and adding a simple classification layer to the reader. Sun et al. (2018) used a common encoding vector between the question and the context to use this vector to verify the candidate answer. However, these methods requires specific adaptation to work with their own reader models while NeurQuRI shows consistent performance improvement when simply combined with a wide range of readers. Similar to our inspection approach, Hu et al. (2019) proposed a verifier network which uses a 12-layer-stacked Transformer with 150M additional model parameters to check answerability of the sentence(s) in which the candidate answer occurs. On the contrary, our method only requires 33M additional model parameters to check answerability of the candidate answer. Additionally, Zhang et al. (2019) proposed a verifier layer which is a linear layer applied to context embedding weighted by start and end distribution over the context words representations concatenated to "[CLS]" token representation for BERT. However, more importantly, our method checks the question's requirements by explicitly comparing question embeddings with the candidate answer embeddings, allowing our model to explain why a question is classified as unanswerable by showing unmet conditions within the question. To this end, unlike Hu et al. (2019) and Zhang et al. (2019), we developed an attention-based satisfaction score that allows our model to reveal which words in the question render it unanswerable.

**Coverage-based Methods and Neural Checklist.** The notion of coverage has been effectively used in various natural language understand and generation tasks. For example, in neural machine translation, Tu et al. (2016) proposed a coverage mechanism that accumulates attention over source text that was already covered while encouraging the model to assign attention on uncovered source text in the subsequent decoding steps. See et al. (2017) leverage this coverage approach in abstractive summarization tasks. Nishida et al. (2019) utilized this coverage mechanism to identify an answer-supporting sentence in multi-hop QA tasks such as HotpotQA (Yang et al., 2018). Nishida et al. (2019) is similar to our method in that it utilize cumulative word-level attention to a given question, but our method has a different goal of checking answerability of a question by inspecting whether conditions from a question are satisfied by a given candidate answer rather than finding answer-supporting sentences. Hence, we newly develop a sophisticated attention module and propose novel loss functions to train our model.

Kiddon et al. (2016) introduced a neural checklist that records the used ingredients of cooking recipes by accumulating attention vectors during the generation of the recipe in a natural language text. They enforce the model to use all the ingredients by the end of the recipe text generation. Although we borrowed the high-level idea from this model, we newly designed the attention accumulation module that is capable of checking multiple conditions simultaneously unlike the softmax approach used in Kiddon et al. (2016). Additionally, we propose the novel recurrent unit to explain answerability and to derive the satisfaction score, which is not addressed in Kiddon et al. (2016).

# 7 CONCLUSIONS

We proposed a novel neural network architecture called Neural Question Requirement Inspector (NeurQuRI), which determines whether the answer candidate generated by a machine reading comprehension model satisfies all the necessary conditions given in the question, in order to determine the answerability of a given question and a context. We evaluated our model on SQuAD 2.0, NewsQA, MS MARCO datasets, which shows consistent performance improvement when combining it with a wide range of existing methods. To demonstrate the effectiveness of NeurQuRI, we also presented an ablation study with respect to different loss terms, as well as the satisfaction score examples computed by NeurQuRI. As long as the question and the answers are encoded with contextual information, we believe that NeurQuRI can be easily extended to other question answering tasks to verify the candidate answer. Future work includes the integration of our approach with an information retrieval system in a way that NeurQuRI properly filters out the retrieved result, together with the end-to-end performance validation of our approach.

**Acknowledgments** We thank all reviewers for valuable and helpful feedback and Sujung Hur for helping in drawing figures. This work was partially supported by Basic Science Research Program through the National Research Foundation of Korea (NRF) funded by the Ministry of Science, ICT & Future Planning (2019R1A2C4070420) and by Korea Electric Power Corporation (Grant number:R18XA05).

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

## A  READER AUXILIARY LOSS

This section explains the auxiliary loss functions that we use to train the reader.

**Normalized Span Loss.** Liu et al. (2018) normalized the span prediction score by adding an empty word to the context input representation. We have a similar normalizing mechanism, but the answerability score of the reader, $z_r$, is computed by using a fully connected unit with two hidden layers that have dimension equal to half the hidden dimension $d$ and a single output dimension. The input of this unit is a concatenation of two vectors, which are attention summations of the final representation over the start index distribution, the end index distribution. We share this answerability score between the start and the end index distributions to make both distributions similarly normalized, i.e.,

$$\mathbb{L}_{r1} = -\log\left(\frac{\phi e^{z_r} + (1-\phi)e^{s_a}}{e^{z_r} + \sum_{i=1}^{n} e^{s_i}}\right) - \log\left(\frac{\phi e^{z_r} + (1-\phi)e^{g_a}}{e^{z_r} + \sum_{i=1}^{n} e^{g_i}}\right),$$

where $s_a$ and $g_a$ represent the scores of the start and the end indices, respectively, of the ground truth answer, $\phi$ indicates the ground truth answerability.

**Independent Span Loss.** Hu et al. (2019) used another span prediction layer separate from the original answer span prediction layer to improve the prediction accuracy of the candidate answer span. In order to allow the reader to extract the candidate answer in unanswerable cases also, they

used plausible answers as the ground truth values for this separate layer. We follow this approach and the loss term for this separate layer $\mathbb{L}_{r2}$ can be written as

$$\mathbb{L}_{r2} = -\log\left(\frac{e^{\tilde{s}_b + \tilde{g}_b}}{\sum_{i=1}^{n}\sum_{j=1}^{n} e^{\tilde{s}_i + \tilde{g}_j}}\right),$$

where $\tilde{s}_b$ and $\tilde{g}_b$ indicate the score of the start and the end indices for the union of a ground truth and a plausible answers. By adding these two losses, the reader auxiliary loss $\mathbb{L}_{aux}$ is written as

$$\mathbb{L}_{aux} = \mathbb{L}_{r1} + \mathbb{L}_{r2}$$

## B    QUALITATIVE ANALYSIS OF FAILED CASES

In this section, we present a few examples from the dev set in SQuAD 2.0 (Rajpurkar et al., 2018) when NeurQuRI incorrectly classifies answerability, as shown in Fig. 4. Consider example 1, an answerable question, in which the question is "Who was one French pro-reform Roman Catholic of the "15th century?". The model could not figure out "15th century" in the questions was referring to "(1455-1536)" in the context, as implied by the low satisfaction scores of "15th century". This possibly points towards a weakness of the contextual embedding supplied to NeurQuRI by the reader.

Example 2 is also an answerable question, "Which direction does two thirds of the Rhine flow outside of Germany?". However, the context does not have any information about Germany, and it does not know that the locations mentioned in the context are "outside" of "Germany", as is evident from their low satisfaction scores. This question comes across as particularly difficult even for humans, requiring extensive real-world geographical knowledge.

In the last example, we show a no-answer question which our model incorrectly classified as answerable with all conditions met. The question "Sir Galileo Galilei corrected the previous misunderstandings about what?", the word "Sir" in the context only refers to "Isaac Newton", and not to "Galileo". This example shows an inherent weakness of our contextualized-embedding based inspection of answerability - the contextualized embeddings also leak information from nearby words, causing the inspection to sometimes pass based on neighbouring words.

Other predecessors of the Reformed church included the pro-reform and Gallican Roman Catholics, such as Jacques Lefevre (c. 1455). The Gallicans briefly achieved independence for the French church, on the principle that the religion of France could not be controlled by the Bishop of Rome, a foreign power. During the ... (omit) ...

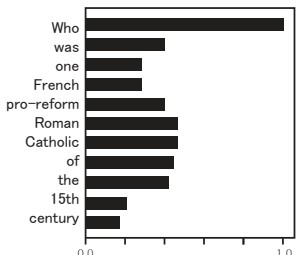

Q: Who was one French pro-reform Roman Catholic of the 15th century?
Predicted → Unanswerable
Ground truth : Answerable

From here, the situation becomes more complicated, as the Dutch name Rijn no longer coincides with the main flow of water. Two thirds of the water flow volume of the Rhine flows farther west, through the Waal and then, via the Merwede and Nieuwe Merwede (De Biesbosch), merging with the Meuse, ... (omit) ...

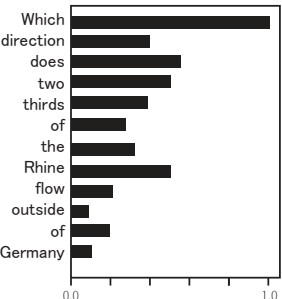

Q: Which direction does two thirds of the Rhine flow outside of Germany?
Predicted → Unanswerable
Ground truth : Answerable

A fundamental error was the belief that a force is required to maintain motion, even at a constant velocity. Most of the previous misunderstandings about motion and force were eventually corrected by Galileo Galilei and Sir Isaac Newton. With his mathematical insight, Sir Isaac Newton formulated ... (omit) ...

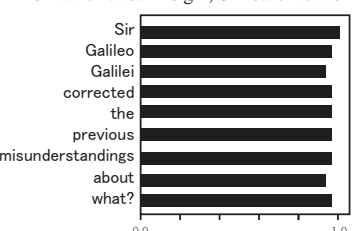

Q: Sir Galileo Galilei corrected the previous misunderstandings about what?
Predicted → Answerable
Ground truth : Unanswerable

Figure 4: Negatively predicted examples of our method, BERT (Large) + NeurQuRI, and its satisfaction score vector $\mathbf{a}_T^{x \to q}$ over question words by the candidate answer. Colored Text indicates the candidate answer predicted by the reader.

