# OpenReview forum: "NeurQuRI: Neural Question Requirement Inspector for Answerability Prediction in Machine Reading Comprehension"
_ICLR.cc/2020/Conference — Accept (Poster)_

### Official Review · AnonReviewer3 · 2019-10-12
**Official Blind Review #3**

**Rating:** 8

**Review:**

The authors propose a Neural Question Requirement module that extracts a list of condition from the question which should be met by the candidate answer in the question answering problems. Authors claim that they propose a novel, attention-based loss function in order to check whether a condition is met.

Evaluation of the methodology is performed on the SquAD 2.0 dataset. It is organized behind combining existing, state of the art solutions like BERT  or QANet. It is improving these solutions consistently.

**Experience Assessment:**

I have published one or two papers in this area.

**Review Assessment: Checking Correctness Of Derivations And Theory:**

I did not assess the derivations or theory.

**Review Assessment: Checking Correctness Of Experiments:**

I assessed the sensibility of the experiments.

**Review Assessment: Thoroughness In Paper Reading:**

I made a quick assessment of this paper.

---

> ### Author Response · Authors · 2019-11-09
> **Response to AnonReviewer3**
>
> We appreciate your compliments!

---

### Official Review · AnonReviewer2 · 2019-10-21
**Official Blind Review #2**

**Rating:** 6

**Review:**


Paper Summary:

This paper proposes a neural question requirement inspection models called NeurQuRI.  It is different from existing answer verifiers in that NeurQuRI pinpoints where the mismatch occurs between the question and the candidate answer in unanswerable cases. Experiments with SQuAD 2.0 show the effectiveness of NeurQuRI.

Strengths:

—NeurQuRI improved the accuracy of three popular reading comprehension models, which are BERT, DocQA and QANet, on SQuAD 2.0.

—NeurQuRI requires 33M additional model parameters to check answerability of the candidate answer.  It is quite less than those of the previous answer verifier (150M) proposed by Hu et al. (2019).

—NeurQuRI can be easily extended to other question answering models and tasks.

Weaknesses:

—The authors do not experimentally compare their model to the answer verifier proposed by Hu et al. (2019). The base model of Hu et al. (2019), RMR, is different from the base models used in this paper, BERT, DocQA, and QANet.

—The authors use only one dataset (SQuAD 2.0) to evaluate their method.  NewsQA, MS MARCO, CoQA and QuAC also contain unanswerable questions.

Comments/Suggestions:

—I think that coverage mechanisms used in NMT (Tu et al., 2016) and summarization (See et al., 2017) should be included in the section of related work.

—The idea similar to NeurQuRI is used in a multi-hop QA model, proposed by Nishida et al. (2019), to find the evidence sentences that cover important information with respect to the question sentence.

—It is worthwhile to present the results of ablation tests with respect to the modified answerability label (described in Section 2.2).

Review Summary:

The paper is well motivated.  However, I think the authors need to compare NeurQuRI with the answer verifier proposed by Hu et al. (2019).  Also, I think this paper can benefit a lot with a more comprehensive analysis with other datasets such as CoQA.

References:
Hu et al.: Read + Verify: Machine Reading Comprehension with Unanswerable Questions. AAAI 2019: 6529-6537
Tu et al.: Modeling Coverage for Neural Machine Translation. ACL (1) 2016
See et al.: Get To The Point: Summarization with Pointer-Generator Networks. ACL (1) 2017: 1073-1083
Nishida et al.: Answering while Summarizing: Multi-task Learning for Multi-hop QA with Evidence Extraction. ACL (1) 2019: 2335-2345

EDIT Nov. 20, 2019:
I appreciate the authors' revision.
The revision has satisfied my concerns, and I decided to increase the score of the paper (weak reject -> weak accept).

**Experience Assessment:**

I have published one or two papers in this area.

**Review Assessment: Checking Correctness Of Derivations And Theory:**

N/A

**Review Assessment: Checking Correctness Of Experiments:**

I assessed the sensibility of the experiments.

**Review Assessment: Thoroughness In Paper Reading:**

I read the paper at least twice and used my best judgement in assessing the paper.

---

> ### Author Response · Authors · 2019-11-09
> **Response to AnonReviewer2**
>
> We appreciate your compliments and valuable feedback.
>
> 1. We will update the manuscript with the F1 score comparison on SQuAD 2.0 development set between NeurQuRI and Verifier from Hu et al. based on DocQA(ELMo) reader as follows,
> DocQA(ELMo)+NeurQuRI - 73.8
> DocQA(ELMo)+Verifier - 70.7. (This result is from Hu et al.,2019)
> NeurQuRI outperforms Hu et. al. verifier by 3.1 F1 score.
> Also note that Hu et. al. unlike our approach does not explain why a question is unanswerable.
>
> 2. We will update the manuscript by evaluating our model on other datasets. We chose SQuAD as our benchmark dataset due to its popularity.
>
> 3. We will update the manuscript of related work with as follows,
> In the case of QFE (Nishida et al. 2019), Hotpot dataset has strong supervision regarding which sentences support/explain the answerability, however NeurQuRI can explain answerability without requiring this supervision, as in datasets such as SQuAD 2.0. We also newly designed the attention accumulation recurrent module that is capable of checking multiple conditions simultaneously unlike the softmax approach used in NMT (Tu et al., 2016),  P-Gen (See et al., 2017), Checklist (Kiddon et al. 2016) and QFE.
>
> 4. We will update the manuscript with an ablation study of the modified answerability label as follows,
> ground-truth label -> Modified label (F1 score)
> BERT+NeurQuRI: 82.6 -> 83.1
> DocQA(ELMo)+NeurQuRI: 72.8 -> 73.8
> QANet+NeurQuRI: 67.5 -> 68.9
> The modified answerability label is more important for weaker readers which are more likely to give an incorrect answer.
>
> References:
> Hu et al.: Read + Verify: Machine Reading Comprehension with Unanswerable Questions. AAAI 2019: 6529-6537
> Tu et al.: Modeling Coverage for Neural Machine Translation. ACL (1) 2016
> See et al.: Get To The Point: Summarization with Pointer-Generator Networks. ACL (1) 2017: 1073-1083
> Nishida et al.: Answering while Summarizing: Multi-task Learning for Multi-hop QA with Evidence Extraction. ACL (1) 2019: 2335-2345

---

> > ### Author Response · Authors · 2019-11-15
> > **Update of submission based on the AnonReviewer2's feedback**
> >
> > Thank you for your constructive feedback and efforts.
> >
> > We have uploaded a new revision with the following changes based on your feedback -
> >
> > 1) Evaluations of our method on NewsQA (Trischler et al. 2017) and MS MARCO (Bajaj et al. 2016) datasets have been added, achieving improvement in both these datasets as well. (Table 2)
> > 2) Comparison with Hu et. al. (2019) answer verifier and SG-Net verifier (Zhang et al. 2019) has been added, and our method outperforms these verifiers. (Table 1)
> > 3) Ablation study with respect to the modified answerability label has been added. (Table 4)
> > 4) Discussion regarding coverage mechanisms and QFE (Nishida et al. 2019) has been added. (Related Works)
> >
> > References:
> > Trischler et al. NewsQA : Newsqa: A machine comprehension dataset. 2017.
> > Bajaj et al. MS MARCO: Ms marco: A human generated machine reading comprehension dataset. 2016
> > Hu et al. 2019: Read + Verify: Machine Reading Comprehension with Unanswerable Questions. AAAI 2019
> > Zhang et al. 2019 : SG-Net: Syntax-Guided Machine Reading Comprehension, https://arxiv.org/abs/1908.05147v1
> > Nishida et al.: Answering while Summarizing: Multi-task Learning for Multi-hop QA with Evidence Extraction. ACL 2019

---

### Official Review · AnonReviewer1 · 2019-10-24
**Official Blind Review #1**

**Rating:** 6

**Review:**

This paper incorporates an answer inspection encoder verifying whether the answers selected by the Machine Reading Comprehension (MRC) component is valid, into the MRC reader. For training, the model includes two additional losses and trained jointly. Evaluation is on SQuAD V2.0 which is constructed from Wikipedia and contains unanswerable questions generated by crowd sources. The approaches are verified in the settings with/without including BERT to show the generalization of the answer inspector.

This is an interesting paper which focuses on the answer verification and validation, and shows the effectiveness of the proposed model. It also gives a good ablation study showing the contributions of each component, and provides examples to illustrate why it works.

However, there are a few concerns detailed as follows:

1. The model is only evaluated on SQuAD 2.0, which is over explored by many works. I’m wondering if this could be generalized to other MRC tasks, e.g. MSMARCO or DuReader. It would be nice to see some experiments on them or other datasets.

2. It seems that the performance of state-of-the-art SOTA system on SQuAD is much higher than the proposed approaches. I would like to see some discussion on what are the pros and cons between them.

3. How sensitive are the gammas in Eq 1?


**Experience Assessment:**

I have published one or two papers in this area.

**Review Assessment: Checking Correctness Of Derivations And Theory:**

N/A

**Review Assessment: Checking Correctness Of Experiments:**

I assessed the sensibility of the experiments.

**Review Assessment: Thoroughness In Paper Reading:**

I read the paper at least twice and used my best judgement in assessing the paper.

---

> ### Author Response · Authors · 2019-11-09
> **Response to AnonReviewer1**
>
> We appreciate your compliments and valuable feedback.
>
> 1. We will update the manuscript by evaluating our model on other datasets. We chose SQuAD as our benchmark dataset due to its popularity.
>
> 2. Our answer inspector can be added in general to any MRC Reader as we have consistently demonstrated using DocQA, QANet and BERT. Below we list the published models in the current SOTA on SQuAD 2.0 leaderboard and compare them to NeurQuRI as follows -
>
> ALBERT - A Lite weight-shared version of BERT. Since we consistently improve BERT’s score with NeurQuRI, we expect performance improvements on applying our answer verifier to ALBERT.
>
> XLNet + SG-Net Verifier - The verifier used in this paper is a linear layer applied to start and end logit weighted token representations concatenated with “[CLS]” token. We reproduce this verifier on our readers, and NeurQuRI outperforms this verifier on SQuAD 2.0 development set as follows (F1 scores) -
> BERT (SG-Net Verifier vs NeurQuRI) - 82.3 vs 83.1
> DocQA(ELMo) (SG-Net Verifier vs NeurQuRI) - 70.7 vs 73.8     [updated. 11/11]
> QANet (SG-Net Verifier vs NeurQuRI) - 67.6 vs 68.9    [updated. 11/11]
>
> RoBERTa - More robustly trained version of BERT. Our verifier can again be directly applied and we expect performance gains.
>
> 3. Regarding the weights of the loss terms in Eq 1, our model is robust to different weights of these loss terms. As we mentioned in Section 3, we use all these relative loss weights set to 1.0. However, other combinations of these weights also lead to comparable scores for all three readers, as long as Lambda_i (corresponding to answerability classification loss) is not made zero as shown in Table 3.
>
> References:
> ALBERT: A Lite BERT for Self-supervised Learning of Language Representations, https://openreview.net/forum?id=H1eA7AEtvS
> XLNET+SG-Net: (Zhang et al. 2019) SG-Net: Syntax-Guided Machine Reading Comprehension, https://arxiv.org/abs/1908.05147v1 [updated. 11/11]

---

> > ### Author Response · Authors · 2019-11-15
> > **Update of submission based on the AnonReviewer1's feedback**
> >
> > Thank you for your constructive feedback and efforts.
> >
> > We have uploaded a new revision with the following changes based on your feedback -
> >
> > 1) Evaluations of our method on NewsQA (Trischler et al. 2017) and MS MARCO (Bajaj et al. 2016) datasets have been added, achieving improvement in both these datasets as well. (Table 2)
> > 2) Comparison with Hu et. al. (2019) answer verifier and SG-Net verifier (Zhang et al. 2019) has been added, and our method outperforms these verifiers. (Table 1)
> >
> > References:
> > Trischler et al. NewsQA : Newsqa: A machine comprehension dataset. 2017.
> > Bajaj et al. MS MARCO: Ms marco: A human generated machine reading comprehension dataset. 2016
> > Hu et al. 2019: Read + Verify: Machine Reading Comprehension with Unanswerable Questions. AAAI 2019
> > Zhang et al. 2019 : SG-Net: Syntax-Guided Machine Reading Comprehension, https://arxiv.org/abs/1908.05147v1

---

### Author Response · Authors · 2019-11-15
**Update of submission based on the reviewers’ feedback**

We would like to thank all reviewers for their constructive feedback and effort in reviewing this paper.

We have uploaded a new revision and would like to summarize the changes we made to the initial submission based on the reviewers’ feedback -

The main changes are -

1) Evaluations of our method on NewsQA (Trischler et al. 2017) and MS MARCO (Bajaj et al. 2016) datasets have been added, achieving improvement in both these datasets as well. (Table 2)
2) Comparison with Hu et. al. (2019) answer verifier and SG-Net verifier (Zhang et al. 2019) has been added, and our method outperforms these verifiers. (Table 1)
3) An ablation study with respect to the modified answerability label has been added. (Table 4)
4) Discussion regarding coverage mechanisms and QFE (Nishida et al. 2019) has been added. (Related Works)

References:
Trischler et al. NewsQA : Newsqa: A machine comprehension dataset. 2017.
Bajaj et al. MS MARCO: Ms marco: A human-generated machine reading comprehension dataset. 2016
Hu et al. 2019: Read + Verify: Machine Reading Comprehension with Unanswerable Questions. AAAI 2019
Zhang et al. 2019 : SG-Net: Syntax-Guided Machine Reading Comprehension, https://arxiv.org/abs/1908.05147v1
Nishida et al.: Answering while Summarizing: Multi-task Learning for Multi-hop QA with Evidence Extraction. ACL 2019

---

### Decision · Program_Chairs · 2019-12-19

**Decision:**

Accept (Poster)

**Comment:**

This paper extracts a list of conditions from the question, each of which should be satisfied by the candidate answer generated by an MRC model. All reviewers agree that this approach is interesting (verification and validation) and experiments are solid. One of the reviewers raised concerns are promptly answered by authors, raising the average score to accept.